# Novel Amphiphilic Polyfluorene-Graft-(Polymethacrylic Acid) Brushes: Synthesis, Conformation, and Self-Assembly

**DOI:** 10.3390/polym13244429

**Published:** 2021-12-17

**Authors:** Maria Simonova, Dmitry Ilgach, Ksenia Kaskevich, Maria Nepomnyashaya, Larisa Litvinova, Alexander Filippov, Alexander Yakimansky

**Affiliations:** Institute of Macromolecular Compounds, The Russian Academy of Sciences, Bolshoy Prospekt 31, 199004 Saint Petersburg, Russia; ilgach@yahoo.com (D.I.); kaskevich-ksenia@yandex.ru (K.K.); marinepom@mail.ru (M.N.); larissa_litvinova@hotmail.com (L.L.); afil@imc.macro.ru (A.F.); alexander.yakimansky@gmail.com (A.Y.)

**Keywords:** amphiphilic polymer brushes, polyfluorene brushes, unimolecular micelles, static and dynamic light scattering

## Abstract

Novel polyfluorene polymer brushes with polymethacrylic acid side chains were obtained by atom transfer radical polymerization (ATRP) and activator generated by electron transfer (AGET) ATRP of tert-butyl methacrylate on polyfluorene multifunctional macroinitiator, followed by protonolysis of the tert-butyl groups of the side chains. Kinetics of polymerization and molecular weights were fully characterized. These polymer brushes luminesce in the blue region of the spectrum with high quantum yields (0.64–0.77). It was shown that the luminescence intensity of polymer brushes is higher than the luminescence intensity of the macroinitiator (0.61). Moreover, due to their amphiphilic nature, they can form unimolecular micelles when an alcohol solution of the polymer brush is injected into water. These properties can potentially be used in drug delivery and bioimaging.

## 1. Introduction

The modern synthetic approaches make it possible to prepare grafted copolymers with a controlled molecular architecture and molar mass. Amphiphilic molecular brushes structure of backbone and side chains which differ strongly have been studied for a long time [1,2,3,4,5,6,7,8,9]. Such brushes attract a significant amount of attention due to the wide possibilities of regulating their characteristics by variation of the chemical structure of components and the architecture parameters, namely, length of main and said chains, and grafting density of the latter. In particular, polymer brushes with a polyimide backbone and side chains of polymethylmethacrylate [10,11,12] or polymethacrylic acid (PMAA) [4,13] have been synthesized and studied in solutions and films. It has been shown that the brushes with PMAA chains may be used as nanocontainers for organic luminophores which are used in photodynamic cancer therapy [13].

In connection with these applications, polymer brushes with conjugated backbone are of interest since they combine luminescence with the unique architecture to expand possibilities of their applications. It is convenient to use polyparaphenylene, polythiophene, and polyfluorene (PF) as such backbones [14,15]. PFs are characterized by high quantum yields of fluorescence, the spectrum of which can be controlled by introducing commercially available phosphors into the chain [16,17]. Moreover, PFs can be modified easily at C9 position for further synthesis of polymer brushes. The luminescent properties of polymer brushes with PF main chain make it possible to obtain materials for sensors, coating for electroluminescence and other applications [18,19,20,21,22,23].

A special direction in the synthesis and research of PF brushes is associated with their application in medicine and biotechnology, in particular for targeted drug delivery [24,25,26,27,28,29,30,31]. Luminescent PF molecular brushes are also called polymer dots, since they have similar properties of quantum dots, but are less toxic, which can be used to diagnose various diseases.

Biomedical applications require good solubility in water, that is ensured by the grafting of water-soluble side chains. In addition, compatibility with water can be achieved by designing supramolecular structures such as micelles, vesicles, polymer nanoparticles, etc., [15,32]. The method of injecting polymer brushes into water is determined by their architectural parameters, first of all the side chain length and the grafting density. In the case of molecular brushes with short side chains and/or low grafting density, micelle-like structures with a hydrophilic shell and a core which are formed due to hydrophobic interactions of the main chains, are usually formed in solutions [4,33,34,35].

The main aim of present work is approbation synthetic ways for preparing of new amphiphilic polymer brushes (APB) with PF backbone and PMAA side chains (Figure 1). Influence of architecture parameters and molar mass characteristics on behavior APB and pre-polymers in solutions and bulk were analyzed also.

In this paper, a method was developed for the synthesis of amphiphilic polymer brushes with a polyfluorene backbone and side chains of polymethacrylic acid, which emit blue light and form micelles in an aqueous solution, which in the future can be used to create materials for targeted drug delivery [36].

## 2. Materials and Methods

2,7-dibromofluorene (97%, Sigma-Aldrich, St. Louis, MI, USA), tetrabutylammoniumbromide (99%, Sigma-Aldrich, St. Louis, MI, USA), 3-bromo-1-propanol (98%, Sigma-Aldrich, St. Louis, MI, USA), sodium hydroxide (99%, Vekton, Russia), 9,9-dioctylfluorene-2,7-diboronic acid bis (1,3-propanediol) ester (97%, Sigma-Aldrich, St. Louis, MI, USA), tetrakis(triphenylphosphine)palladium(0) (Pd(0)[PPh_3_]_4_ (99.99%, Sigma-Aldrich, St. Louis, MI, USA), triphenylphosphine (99%, Sigma-Aldrich, St. Louis, MI, USA), tricaprylmethylammonium chloride (97%, Aliquat^®^ 336, Sigma-Aldrich, St. Louis, MI, USA), potassium carbonate (99%, Sigma-Aldrich, St. Louis, MI, USA), phenylboronic acid pinacoline ester (97%, Sigma-Aldrich, St. Louis, MI, USA), p-bromoethoxy benzene (98%, Sigma-Aldrich, St. Louis, MI, USA), 2-bromoisobutyryl bromide (98%, Acros Organics, NJ, USA), copper(II) chloride (99%, Acros Organics, NJ, USA), 4,4′-dinonyl-2,2′-bipyridyl (97%, Sigma-Aldrich, St. Louis, MI, USA), copper(I) chloride (99%, Acros Organics, NJ, USA), tin(II) 2-ethylhexanoate (95%, Alfa-Aesar, Ward Hill, MA, USA), potassium hydroxide (99%, Vekton, Russia). Anisole (99%, Sigma-Aldrich, St. Louis, MI, USA) was distilled over sodium (Na) twice. Tetrahydrofuran (THF, 99%, Vekton, Russia) was distilled over calcium hydride. *Tert*-butyl methacrylate (TBMA, 98%, Sigma-Aldrich, St. Louis, MI, USA) was distilled in vacuum. Dimethylsulfoxide (98%, Vekton, Russia), methanol (98%, Vekton, Russia), triethylamine (99%, Sigma-Aldrich, St. Louis, MI, USA), methylene chloride (98%, Vekton, Russia), trifluoroacetic acid (99%, Sigma-Aldrich, St. Louis, MI, USA) were used without further purification. Quinine sulfate, a standard for the quantum yield measuring, was purchased from ThermoFisher Scientific (Reference Dye Sampler Kit (R-14782)).

NMR spectra were recorded on a Bruker AVANCE–400 SB (400 MHz). The UV-visible absorption spectra were recorded on a Shimadzu UV-1900 spectrophotometer. Photoluminescent spectra were measured using a RF-6000 spectrofluorophotometer. The kinetic of reaction was studied using a Shimadzu GC-2010Plus gas chromatograph. FT-IR spectra were recorded on a Shimadzu IRAffinity-1S using a Quest single ATR attachment (Specac), diamond prism, 7800–400 cm^−1^ range. Thermogravimetric analysis (TGA) was conducted on TG 209 F1 Libra from Netzsch by heating the samples (2 mg) to 800 at a rate of 10/min in a nitrogen atmosphere. Differential scanning calorimetry (DSC) was investigated on DSC 204 F1 Phoenix from Netzsch by heating samples from 25 to 150 °C two times at a rate of 10 °C/min in a nitrogen atmosphere.

### 2.1. Synthesis of 2.7-Dibromo-9,9-Bis(3-Hydroxylpropyl)-Fluorene (***1***)

2,7-dibromofluorene (3.86 g, 11.9 mmol) and tetrabutylammoniumbromide (0.76 g, 2.3 mmol) were dissolved in dimethylsulfoxide (60 mL) in a 100 mL three-neck round-bottom flask. Next, 10 mL of a 50% NaOH solution was injected under argon. The solution was stirred at room temperature for 30 min. Then 3-bromo-1-propanol (4 mL, 45.2 mmol) was added dropwise into the flask for over 40 min. The resulting mixture was warmed to 85 °C under constant stirring for 24 h. Then it was poured into water, extracted with dichloromethane, and the dried over Na_2_SO_4_, followed by filtration. Solvent was removed and the product was purified by recrystallization from chloroform two or three times. The final product was a light-yellow solid. Yield: 70%. ^1^H NMR (400 MHz, DMSO-*d_6_*, δ): 7.83 (d, J = 8.1 Hz, 2H, Ar H), 7.69 (d, J = 1.8 Hz, 2H, Ar H), 7.56 (dd, J = 8.1, 1.8 Hz, 2H, Ar H), 4.23 (t, J = 5.3 Hz, 2H, 2OH), 3.13 (q, J = 6.6 Hz, 4H, 2CH_2_), 2.07–1.98 (m, 4H, 2CH_2_), 0.69–0.60 (m, 4H, 2CH_2_). ^13^C NMR (101 MHz, DMSO-*d_6_,* δ): 152.79, 139.28, 130.61, 126.61, 122.52, 121.50, 61.25, 55.52, 35.86, 27.76.

### 2.2. Synthesis of Poly[(9,9-Bis(3-Hydroxylpropyl)-Fluorene)-Alt-(9,9-Dioctylfluorene)] (***P1***)

9,9-dioctylfluorene-2,7-diboronic acid bis (1,3-propanediol) ester (0.5704 g, 0.001 mol), 3,3′-(2,7-dibromo-9*H*-fluorene-9,9-diyl) dipropan-1-ol (0.4400 g, 0.001 mol), triphenylphosphine (5 mg) were added to a glass round bottom flask. The flask was closed with a reflux condenser and an adapter with a sampling tap and a vacuum valve (Aldrich Z530255). Then the system was evacuated and filled with argon three times. The catalyst-Pd(0)[PPh_3_]_4_, was weighted (5 mg) in a glove box under argon atmosphere and added to the flask. Then reaction system was evacuated and filled with argon three times again.

Solution of Aliquat^®^336 (20 mg) in anisole (1.5 mL), anisole (14.5 mL), and 2M K_2_CO_3_ in bidistilled water (10 mL) were introduced into the system using a syringe through a septum in an adapter. The reaction was carried out in a CEM Discover SP microwave reactor at a temperature of 100 °C and a power of 80 W (Mode: SPS mode, ΔT = 5 °C) for 90 min. To close polymers end groups, a solution of phenylboronic acid pinacoline ester (0.012 g, 3 mol%) in anisole (1 mL) was added to the system using a syringe, and the reaction was continued at 100 °C for an hour. Then p-bromoethoxy benzene (200 μL, 3 mol%) in anisole (1 mL) was added to the mixture, and the reaction was carried out for another 90 min.

Solutions and solvents were purged with argon for an hour before being added to the system.

The polymer solution was diluted with THF, was separated from water solution, dried from water over sodium sulfate and passed through a layer of modified silica gel (QuadraSil MP, 20–100 micron, metal scavenger). Then the solution was concentrated and precipitated into methyl alcohol. The polymer was filtered off, washed with methanol four times, with water, again with methanol and dried in vacuum at 50 °C. After that the polymer was reprecipitated from THF to methanol and isolated in a same way. Yield: 60%.^1^H NMR (400 MHz, Chloroform-*d*, δ) 7.92–7.56 (br, 12H, Ar H), 3.49 (br, 2H, 2OH), 2.23 (br, J = 55.1 Hz, 8H, 4CH_2_), 1.37–1.02 (m, 28H, 14CH_2_), 0.95–0.76 (m, 6H, 2CH_3_).

### 2.3. Synthesis of Macroinitiator MI

**P1** (0.1 g) and triethylamine (1.41 mL, 10.1 mmol) were dissolved in 24 mL anhydrous THF. The mixture was stirred in an N_2_ atmosphere and cooled to 0 °C. Next, 2-bromoisobutyryl bromide (0.96 mL, 7.6 mmol) was added dropwise for 20 min. The mixture was warmed to room temperature and kept under constant stirring for 24 h. Triethylamine hydrobromide salt was filtered off. The resulting solution was concentrated and precipitated into methanol. The final product was dried in a vacuum resulting in a light-yellow solid. Yield: 75%. The sample was characterized by SEC in THF using a light scattering detector to obtain the molecular weight (M_n_ = 21,030 g·mol^−1^, M_w_/M_n_ = 1,5). ^1^H NMR (400 MHz, Chloroform-*d*, δ) 8.13–7.51 (br, 12H, Ar H), 4.04 (br, J = 5.9 Hz, 4H, 2CH_2_), 2.43–2.03 (br, 8H, 4CH_2_), 1.88 (s, 12H, 4CH_3_), 1.31–1.01 (m, 28H, 14CH_2_), 0.96–0.71 (m, 6H, 2CH_3_).

### 2.4. Synthesis of Polymer Brush with Polyfluorene Backbone and Poly-Tert-Butyl Methacrylate Said Chains (PB)

*ATRP mechanism*. MI (0.1 g, 0.084 mmol), CuCl_2_ (1.1 mg, 0.0084 mmol), 4,4-dinonyl-2,2-bipyridyl (dNbpy, 0.206 g, 0.504 mmol) were placed into a Schlenk flask and were dissolved in anisole (2.72 mL, 33vol%). Then TBMA (1.36 mL, 8,4 mmol) was added. Flask was closed with rubber septum and the reaction mass was degassed by three freeze-pump-thaw cycles. After that CuCl (16.6 mg, 0.168 mol) was added on the ice under argon. Flask was evacuated and filled with argon. The reaction was carried out on a magnetic stirrer at 80 °C. Anisole and TBMA were purged with argon for one hour before adding to the system.

At the end of the reaction, the mixture was cooled and diluted with THF. Next the solution was purified from the catalyst by column chromatography on aluminum oxide (activated, neutral). The resulting solution was evaporated and precipitated into a mixture of MeOH/water (10/1, *v*/*v*). The precipitate was filtered and washed with the same mixture. The resulting product was dried under vacuum.

*ATRP AGET mechanism*. MI (0.072 g, 0.061 mmol), CuCl_2_ (16 mg, 0.12 mmol), dNbpy (0.149 g, 0.36 mmol) were introduced into a Schlenk flask. The flask was closed with a septum, evacuated and backfilled with argon three times. Anisole (8 mL, 33 vol%) was added through the septum. TBMA (4 mL, 0.0244 mmol) was added after the dissolution of the components. Then an aliquot of a solution of Sn(EH)_2_ (0.049 g, 0.12 mmol) in anisole (1 mL) was added to the reaction mixture through the septum. The reaction was carried out under continuous stirring at 80 °C. Anisole, TBMA, and Sn(EH)_2_ solution were purged with argon for 30 min before adding to the system.

At the end of the reaction, the mixture was cooled and quenched with THF. Polymer was isolated at the same time, as described above.

Determination of monomer conversion: aliquots (10 μL) were taken from the reaction mixture and diluted in THF (200 μL) and analyzed by gas chromatography. Anisole was used as an internal standard.

Agilent Technologies 1260 Infinity liquid chromatograph (The Agilent 1260 Infinity Multi-Detector GPC/SEC System) equipped with three detectors: refractometric (DRI, Laser wavelength 660 nm), viscometric (VS), and light scattering (LS: Rayleigh scattering angles 15° and 90°; Laser wavelength/power: 660 nm/50 mW) was used for the determination of molar masses M_w_ and heterogeneity index or molar-mass dispersity *M*_w_/*M*_n_ of investigated polymers by size exclusion chromatography (SEC). The setup included a set of chromatographic columns connected in series: a PLgel 5 μm Guard 50 × 7.5 mm precolumn and two Agilent Technologies PLgel 5 μm MIXED-C, 300 × 7.5 mm columns. The temperature of detectors and columns was 40 °C. THF was used as mobile phase with the rate 1.0 mL/min. The concentration of injected sample did not exceed 2 mg/mL. PS-standards (Agilent Technologies) were used for columns calibration in GPC regime (DRI detector).

Hydrolysis of PBs was carried out under the conditions listed below for characterization of molar masses of the said chains: PB (60 mg) was dissolved in THF (8.8 mL) and of 5% KOH solution in MeOH (16 mL). The reaction was carried out at 90 °C for a day. At the end of the reaction, the solution was evaporated and washed with water.

### 2.5. Synthesis of Amphiphilic Polymer Brush with Polyfluorene Backbone and Poly Methacrylic Acid Said Chains (APB) and Preparation of APB Nanoparticles

PB (0.1 g) was dissolved in methylene chloride (2.5 mL) in a 25 mL round bottom flask and CF_3_COOH (0.8 g) was added. The reaction was carried out on a magnetic stirrer at room temperature for a day. Then mixture was exposed to air and the solvents were removed. The powder was dried in vacuum at 50 °C for 4 h.

To obtain nanoparticles APB (3 mg) was dissolved in ethanol (0.95 mL). Then the solution (0.5 mL) was injected into distilled water (2.5 mL) under sonication.

### 2.6. Determination of Molar Masses, Hydrodynamic Characteristics and Investigation of Self-Assembly of Molecular Brushes in Dilute Solutions

Static (SLS) and dynamic (DLS) light scattering were used to study the solution behavior of copolymers with different compositions by Photocor Complex instrument (Photocor Instruments Inc., Moscow, Russia). The light source was the Photocor-DL diode laser with the wavelength λ = 659.1 nm and controllable power up to 30 mW. Procedure was described early [33].

As is known, the behavior of amphiphilic polymers in solutions strongly depends on the thermodynamic quality of the solvent with respect to the components. Therefore, the choice of solvents for research is an important task. As can be seen from Table 1, the blocks of the considered brushes (MI, PtBMA, PMAA) dissolve in different ways in the solvents we have chosen.

The synthesized precursors and molecular brushes (PB, APB), were molecularly dissolved in chloroform. Therefore, their molar mass characteristics were determined in chloroform (Sigma-Aldrich, Saint Louis, MO, USA)

The molar mass of the PF macroinitiator were measured in chloroform, which is a thermodynamically good solvent for PF.

## 3. Results and Discussion

### 3.1. Synthes of Polymer Brushes

A multicenter polyfluorene macroinitiator (MI) was obtained in several stages (Figure 2). First, **P1** with side hydroxyl groups poly-9,9-bis(3-hydroxypropyl)-alt-9,9-bis(dioctyl) fluorene was obtained by the Suzuki reaction. If the reaction was carried out in toluene, the polymer solution was cloudy, as described in the literature [18]. Therefore, a more polar solvent was used, namely anisole, in which the reaction mix was transparent. In both toluene and anisole, cooling solution to room temperature led to precipitation of PF, due to probably the formation of interchain hydrogen bonds between hydroxyl groups. It was also shown that carrying out the synthesis in a microwave reactor makes it possible to obtain PF in 4.5 h, instead of 72 h under normal conditions [18].

Second, MI was prepared by modification of the side hydroxyl groups of polyfluorene with 2-bromisobutyroyl bromide, using a well-known method (Figure 2) [18]. The degree of functionalization of the MI, i.e., the molar fraction of reacted hydroxyl groups, was determined using ^1^H NMR spectroscopy by the ratio of the integral intensities of aromatic protons and protons of bromoether groups (Appendix A). Functionalization degree of MI was 75%.

Polymer brushes polyfluorene-graft-poly-tert-butyl methacrylate (PB) were synthesized by the «grafting from» method by means of tert-butyl methacrylate polymerization on MI. Two polymerization mechanisms were used: ATRP and AGET ATRP. The synthesis was carried out in anisole, which is also used as an internal standard for determining the monomer conversion by gas chromatography. The molar ratio of the initiating group/monomer and the synthesis time were varied in order to obtain polymer brushes with different lengths of side chains. Synthetic conditions of PBs are shown in Table 2. It was found that ARGET method was more convenient and simpler than normal ATRP and help us to obtain polymer brushes in a controlled manner.

Kinetic of polymerization was studied by gas chromatography (samples PB5-PB8). Figure 3 shows a graph of the kinetics, which begins to bend after 50% conversion, which may indicate crosslinking of polymers during synthesis.

Finally, APBs were prepared by deprotection the tert-butyl deprotection from the side chains of PBs with trifluoroacetic acid (Figure 2). The completeness of the passage was confirmed by IR and NMR spectroscopy (ESM, Appendix A).

Figure 4 shows that the chromatograms were unimodal. The polydispersity does not increase much when side chains are grafted to polyfluorene. This fact indicates the successful synthesis of polymer brushes and indirectly reflects of the relatively narrow polydispersity of the side chains. For a more accurate analysis of the molecular weight characteristics of the side chains, the polymer brush was hydrolyzed along the ester bond connecting the main and side chains in order to isolate the poly-tert-butyl methacrylate side chains. The previously developed conditions of hydrolysis in a similar system were used, which allow hydrolysis of the ester group without affecting the tert-butyl groups of the side chains [34].

Static and dynamic light scattering in dilute solutions were used for determination of the molar mass and hydrodynamic characteristics of synthesized brushes and investigation of self-assembly.

It was found that APB do not dissolve in water, but dissolve in alcohol. However, if an alcohol solution of APB is injected into water, micelles are formed, the properties of which will be described below.

Studies of solutions of the multicenter macroinitiator and PB in chloroform and APB in methanol were carried out. Using photoluminescence spectra, it was shown that the luminescence intensity of polymer brushes is higher than that of the macroinitiator, despite the fact that the mass fraction of the luminescent polymer is higher in the macroinitiator compared to polymer brushes in which only the main chain luminesces (Figure 5). In this case, the most absorption is observed in the multicenter macroinitiator, since it consists of conjugated condensed polyfluorene rings. After grafting of the side chains, the mass fraction of the main chain in the polymer brushes became smaller, which leads to a decrease in the absorption intensity.

The above results are confirmed by the data of quantum yields (QY) of polymer brushes PB (Table 3) and APB (Table 4), which were determined by a known method [37]. Quinine sulfate in 1.0 N sulfuric acid solution was used as a standard, the quantum yield of which is 0.55 [38]. The results presented in Table 3 and Table 4 show that the growth of side chains in the polymer brush leads to an increase in the quantum yield relative to MI (0,61), and a decrease in the quantum yield of the APB (0.64–0.72) brushes relative to the PB (0.64–0.77) is observed. The increase in the quantum yield is probably associated with the destruction of aggregates with an increase in the length of the side chains in the case of PB, while the decrease in the quantum yield of amphiphilic polymer brushes occurs due to the formation of aggregates between the side chains of poly methacrylic acid due to the formed hydrogen bonds. Absorption and luminescence spectra of polymer brushes PB and APB can be found in ESM Appendix A.

### 3.2. Molecular Hydrodynamic Characteristics of MI, PB, and APB

Molar masses of macroinitiator were determined in THF. As can be seen in Figure 4, the chromatograms in this solvent were unimodal, the weight-average molar masse *M*_w_ was 31,000 g/mol with a low polydispersity for polycondensation polymers. Accordingly, weight-average the polymerization degree *N*_MI_ of the macroinitiator is 36, and the contour length *L*_MI_ of its chain is 54 nm, taking into account that the length of the MI monomer unit is 1.5 nm. Since the degree of functionalization *z** = 0.75, each PF molecule contains, on average, 27 functional groups. Note that the solutions of the macroinitiator in chloroform were not molecularly dispersed. Bimodal distribution was observed in this solvent by dynamic light scattering (ESM, Appendix A). In addition to macromolecules, aggregates were present in the solution, which may explain the grafting degree and presents hydroxyl groups.

In opposite, the chloroform solutions of the copolymer PB were molecularly dispersed. There was no concentration dependence of hydrodynamic radius *R*_h_(*c*) measured at concentration *c* for most of the samples (ESM, Appendix A), and therefore the concentration-averaged *R*_h_(*c*) value was taken as the hydrodynamic radius *R*_h-D_ of macromolecules. Molar masses and hydrodynamic characteristics are presented in Table 3.

It is interesting to compare molar masses of brushes obtained by SLS and calculated using structure parameters and synthesis conditions. Note that using of number-average molar masses simplifies calculations, however, each time it is necessary to take into account the polydispersity of the samples when comparing the obtained MMs with the experimental results. Consequently, the number-average molar mass *M*_n_^cal^ of the PB copolymers can be expressed as:*M*_n_^cal^ = *N*_MI_ × *z* × *M*_s.ch._ + *M*_MI_,(1)
where *z* is the graft density, or ratio of number of side chain to polymerization degree of MI, and *M*_s.ch_ is the calculated MM side chain. In the calculations, it was assumed that *z* = *z**.

Table 3 lists the *M*_w_^cal^/*M*_w_^exp^ ratio, i.e., compares the experimental *M*_w_^exp^ and calculated *M*_w_^cal^ molar masses for the investigated molecular brushes. It is clearly seen that the values of *M*_w_^cal^/*M*_w_^exp^ are close to 1 for grafted copolymers with relatively shorter side chains (expected polymerization degree of side chains *N*_s.ch._ = 50). Therefore, during the synthesis of these polymers, the side chains are polymerized from each functional group, i.e., *z* is equal to *z**. With lengthening of the side chains, the difference between *M_w_*^exp^ and *M*_w_^cal^ is observed, the greater *M*_w_^cal^/*M*_w_^exp^ the greater the MM of side chains. Accordingly, it can be assumed that with an increase in *N*_s.ch_, either brushes are formed, the side chains of which are shorter than expected based on the synthesis conditions, or grafted copolymers with a lower grafting density (*z* < *z**) are prepared.

An important structural parameter of molecular brushes is the ratio *L*_s.ch._/Δ*L* of the side chain length *L*_s.ch._ to the distance Δ*L* between the grafting points. The *L*_s.ch._/Δ*L* value characterizes steric side chain interactions: the more *L*_s.ch._/Δ*L*, the stronger these interactions. In a thermodynamically good solvent, strong interactions provide an extended conformation of the side chains. In a selective solvent, long side chains at a small distance Δ*L* are good at shielding the backbone from the solvent. The value of *L*_s.ch._/Δ*L* is easy to calculate by the formula
*L*_s.ch._/Δ*L =* (*M*_w_^exp^ − *M*_MI_)/*L*_MI_ *M*_L-s.ch._(2)
where *M*_L-s.ch._ = *M*_0-s.ch./_λ_0-s.ch._ is the molar mass per unit length of the side chain and *M*_0-s.ch._ and λ_0-s.ch._ are the molar mass and length of monomer unit, respectively. For PtBMA, *M*_0-s.ch._ = 142 g·mol^−1^ and λ_0-s.ch._ = 0.252 nm. The values of *L*_s.ch._/Δ*L* for investigated grafted copolymer are presented in Table 3. It is clearly seen that *L*_s.ch._/Δ*L* lie in the range from 8 to 18, that is, for all samples PB, the side chains are long enough. It determines their strong steric interactions and good shielding of the backbone from a solvent.

In order to describe the conformation and hydrodynamic behavior of PB, the Porod wormlike chain model and the wormlike spherocylinder model [39,40,41] were used. Choosing the molecular model, we kept in mind the structure of samples under investigation, namely, sufficiently long main PF and side PtBMA chains, and also the fact that the ratio is large enough. This approach was successfully applied to determine the molecular parameters of cylindrical brushes [42,43,44,45,46,47,48,49].

In the general case, the dependences of intrinsic viscosity [η] and hydrodynamic radius *R*_h-D_ on molar mass *M*, contour length *L* of model spherocylinder, its diameter *d*, and Kuhn segment length A are described by the equations:[η] = Φ(*L*/*A*)^3/2^/*M*,(3)
3*π*η_0_*L*/*f* = 2*L*/*R*_h_ = Ψ,(4)
where *f* is the translational friction coefficient of macromolecules, Φ and Ψ are functions of the aforementioned parameters. The forms of functions Φ and Ψ are different for short (*L*/*A* ˂ 2.278) and long molecules. Exact values of Φ(*L*, *d*, *A*) and Ψ(*L*, *d*, *A*) have been defined and described [39,41].

Using equations 3 and 4 makes it possible to estimate the molecular parameters *L*, *d*, and *A* from the experimental values of [η] and *R*_h-D_. In order to find *L*, *d*, and *A* we varied their values achieving simultaneous solution of Equations (3) and (4) [50]. The obtained values of *d* and *A* are listed in Table 3.

For all samples under study, the Kuhn segment length is slightly larger than *A* = 15 nm for polyfluorene. The change in the *A* value in the series of the PB samples is small. It is caused by probably a low difference in the values of the ratio *L*_s.ch._/Δ*L*, an increase in which leads to an increase in the Kuhn segment length. For example, for sample PB3 *A* = 17–19 nm at *L*_s.ch._/Δ*L* = 9.5, and for PB7 *A* = 19–22 nm at *L*_s.ch._/Δ*L* = 18. The insignificant difference in the equilibrium rigidity of the studied copolymers from that for PF can be explained by the high rigidity of the latter. Indeed, in the case of a rigid-chain backbone, the presence of side chains does not usually have a very strong effect on the Kuhn segment length [51]. The effect of the structure is somewhat more noticeable for the values of the hydrodynamic diameter *d*. On passage from samples with *L*_s.ch._/Δ*L* = 8–11 to samples with *L*_s.ch._/Δ*L* = 17–18, the *d* value increases by about 30–50 percent. This fact makes it possible to assume that the length of the side chains of copolymers with a higher *L*_s.ch._/Δ*L* value is greater than for all other samples. It should be noted that for all samples the minimum contour length of the side chains exceeds the obtained values of the radius *d*/2 of the modeling spherocylinder. Therefore, we can conclude that the grafted chains are folded.

An important characteristic of the macromolecule behavior in solutions is the so-called hydrodynamic invariant *A*_0_ [51,52,53], values of which may be calculated using experimental values of molar mass *M*_w_, intrinsic viscosity [η], and diffusion constant by the formula.
*D*_0_ = *kT*/(6πη_0_*R*_h_),(5)
*A*_0_ = η_0_*D*_0_(*M*[η]/100)^1/3^/*T*,(6)

The *A*_0_ value depends on conformation and architecture of macromolecules. For example, the average experimental values of hydrodynamic invariant for flexible-chain polymers is 3.2·10^−10^ erg·K^−1^mol^−1/3^ and for rigid chain polymers is 3.8·10^−10^ erg·K^−1^mol^−1/3^ [51,52,53].

The values of *A*_0_ for solutions of investigated brushes in ethanol are listed in Table 3. The dispersion is large, however there is no systematic change in the hydrodynamic invariant from molar mass or architectural parameters. The average value is equal to *A*_0_ = (2.0 ± 0.4)·10^−10^ erg·K^−1^mol^−1/3^. It is close to hydrodynamic invariant for hyperbranched polymers, such as polyamino acids, perfluorinated polyphenylenegermanes, and polycarbosilanes [54,55,56]. It can be assumed that, as in the case of hyperbranched polymers, the low values of the invariant for solutions of the studied molecular brushes are caused by their high intramolecular density. Note that for a grafted copolymer with a polyimide backbone and side chains of polymethylmethacrylate *A*_0_ = 2.4·10^−10^ erg·K^−1^mol^−1/3^ is also lower than the values for linear polymers (this value of *A*_0_ was calculated by us from the experimental values of hydrodynamic parameters given in the article [57]).

For APB in ethanol solution the distribution of intensity *I* over the hydrodynamic radii *R*_h_ of scattering objects was unimodal (Figure 6). The particle radius *R*_h_ was independent of the concentration (ESM, Appendix A) and its average value was slightly different from the hydrodynamic size *R*_h-D_ of molecules of PB in chloroform. On the other hand, for each sample the intrinsic viscosity of APB in ethanol is slightly higher than [η] of PB in chloroform. Although this difference is small, it may indicate an increase in the size of dissolved objects in APB ethanol solution compared with PB chloroform solution.

Static light scattering provided direct information about the particle size. As indicated above, the asymmetry of light scattering for all solutions of APB in ethanol was observed, and the Zimm method was used to analyze the light scattering data (ESM, Appendix A). This made it possible to determine not only the MM values, but also the gyration radii R*_g_* of the dissolved objects (Table 4).

Comparison of the data in the Table 3 and Table 4 shows that the molar masses of PB determined in chloroform are lower than the MM of APB obtained in ethanol. Taking into account the difference in the molar masses of the monomer unit of PtBMA and PMAA, one would expect a decrease in MM at passage from PB to APB by about 1.6 times. Together with the viscometry data, this fact makes it possible to assume that aggregates are formed in ethanol. Their sizes are small; based on the values of the molar-masses, the aggregation degree *m*_a_ varies from 2 to 10. No dependence of *m*_a_ on the expected length of the side chains can be traced.

Some conclusions about the aggregate shape can be made based on the values of the so-called form factor, namely, the ratio *R_g_*/*R*_h_ of the gyration radius *R_g_* to hydrodynamic radius *R*_h_. For most of the investigated samples the *R_g_*/*R*_h_ values are higher than 2. Such high *R_g_*/*R*_h_ are typical for long particles, such as a cylinder or an elongated revolution ellipsoid.

As mentioned above, it was possible to obtain aqueous solutions of APB. Unfortunately, the method of obtaining these solutions does not allow reliable determination of their concentration. However, taking into account that the intensity of light scattering by aqueous solutions of APB is much less than *I* of solutions in ethanol, we can assume a sharp decrease in the concentration of the polymer in water as compared with one in ethanol. This does not allow carrying out the complex studies in water (determination of MM, viscometry, refractometry, etc.). Therefore, we can only make preliminary conclusions based on the values of the hydrodynamic radius, measured reliably. As can be seen from Table 4. the *R*_h_ value in water is not very different from that observed in ethanol. Accordingly, it can be assumed that solutions in water are also nonmolecular with an aggregate size close to that in ethanol solutions.

### 3.3. Study of the Thermal Characteristics of Polymer Brushes

Macroinitiators and polymer brushes with lateral PtBMA were analyzed by thermogravimetry. Three regions can be distinguished on the decomposition curve of the macroinitiator (Figure 7). The first section of the curve −200–300 °C (weight loss 14%) corresponds to the elimination of bromine atoms, the second −300–420 °C (weight loss 26%) corresponds to the destruction of ester bonds, the third −420–555 °C corresponds to the elimination of alkyl substituents from polyfluorene (weight loss 18%). The residual mass −42%, is obviously the carbonized residue of the polyconjugated polyfluorene system. The decomposition curves for linear PtBMA and polymer brush are similar. They consist of two stages: the first is the destruction of ester bonds with the elimination of tert-butyl groups in the range 200–320 °C (weight loss PB2 −42%, PB5 −46%), 190–350 °C (weight loss of linear PtBMA −48.7%), destruction of the carbon chain polymer up to 500 °C, which is accompanied by the elimination of alkyl substituents from the polyfluorene in the polymer brush. It can be seen that linear PtBMA decomposes almost completely, whereas after the decomposition of the polymer brush, there is a residue of the carbonized backbone (6.01% by weight).

PB were investigated by DSC. The glass transition temperature (*T*_g_) of polymer brushes is higher than that of linear PtBMA (98.7 °C) and lies in the range of 113.9–119.5 °C. Moreover, the higher the MM of the polymer brush, the higher the *T*_g_.

## 4. Conclusions

The series of amphiphilic molecular brushes with the polyfluorine backbone and PtBMA and PMAA side chains were synthesized by radical polymerization. The structure and composition of synthesized molecular brushes were confirmed by comparing molar masses of copolymers and their components. Solution behavior MI of PB and APB samples were determined by their structural parameters and nature of solvent. In molecular disperse solutions (in chloroform), the molecules of PB of the synthesized graft-copolymers had compact sizes.

The size obtained for the APB micelles in water coincides with the experimental error of the ones of APB in ethanol. This fact confirms stable macromolecular structure. Unfortunately, we can conclude the form of structures, but we could not determine the precise concentration. Nevertheless, it is clearly seen that *R_g_*/*R*_h_ is higher than 2.

Amphiphilic luminescent polyfluorene brushes APB form micelles in water, which in the future can be used to create materials for targeted drug delivery. For example, we found that such micelles can incorporate hydrophobic model compound–curcumin, forming stable colloid solutions. Moreover, we are going to solubilize hydrophobic potential photosensitizer [tetra(4-fluorophenyl)tetracyanoporhyrazine free base] by APB-like nanocontainers for photodynamic therapy [13]. Mentioned photosensitizer emits light at 650–710 nm whereas APB emits blue light at 400–475 nm. This probably allows to visualize the mechanism of distribution of these objects in living cells.

## Figures and Tables

**Figure 1 polymers-13-04429-f001:**
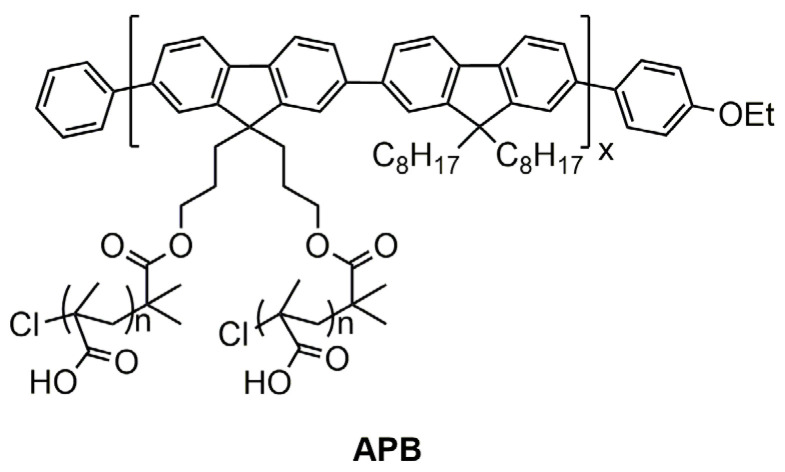
Structure of amphiphilic polymer brushes with a polyfluorene backbone and polymethacrylic acid side chains.

**Figure 2 polymers-13-04429-f002:**
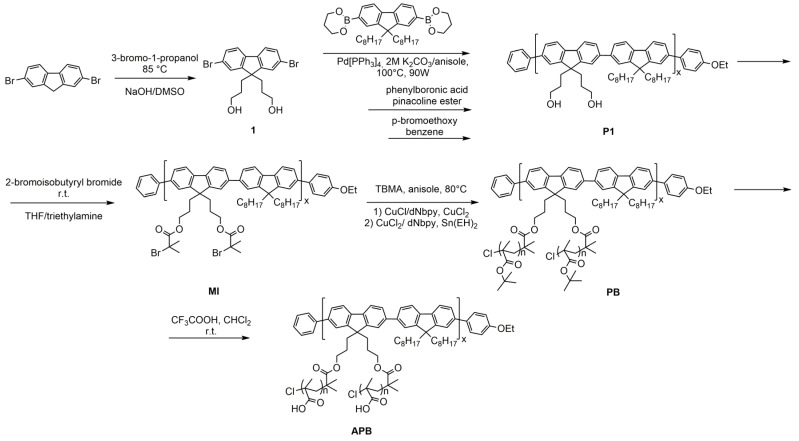
Synthesis of polymer brushes.

**Figure 3 polymers-13-04429-f003:**
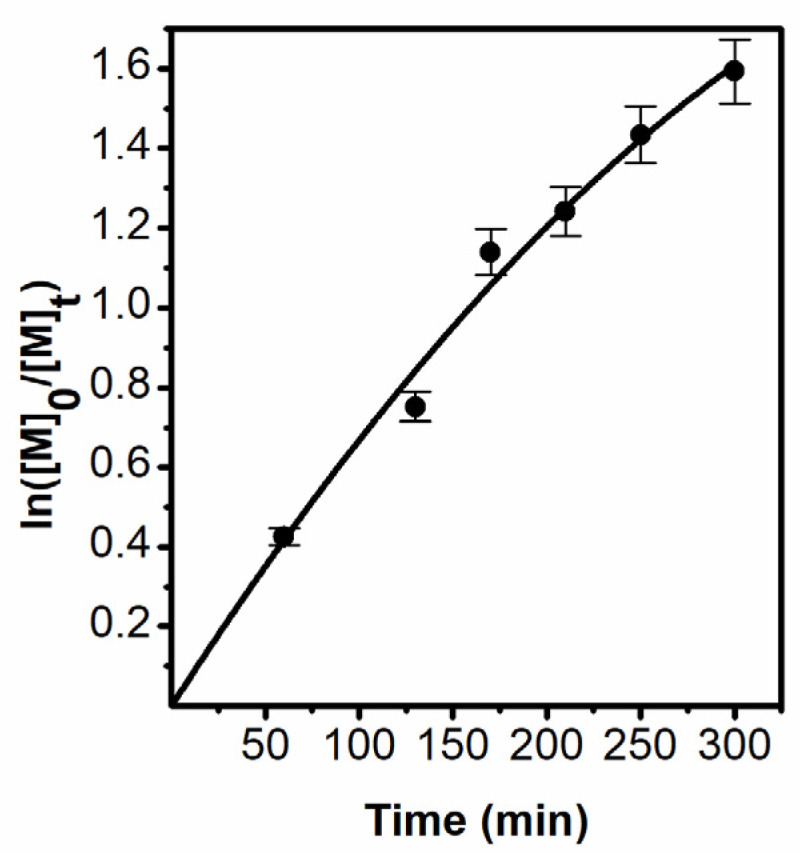
ln([M]_0_/[M]_t_) plot vs. time in solution polymerization (33 vol%) of TBMA in anisole from AGET mechanism, molar ratio [MI]:[CuCl_2_]:[Sn(EH)_2_]:[dNbpy] = 1:1:1:3.

**Figure 4 polymers-13-04429-f004:**
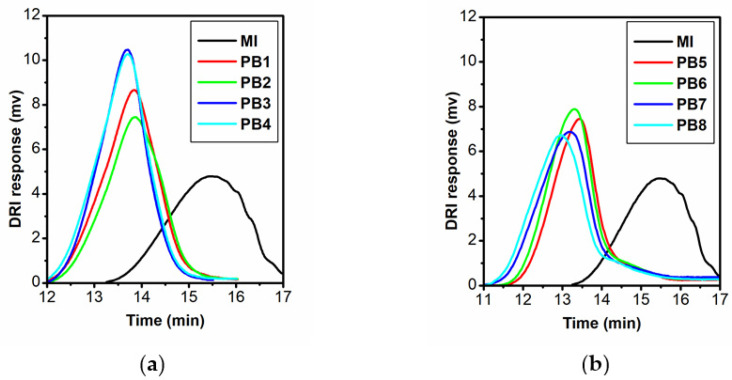
SEC/DRI chromatograms obtained for: (**a**) MI and PB1-PB4; (**b**) MI and PB5-PB8.

**Figure 5 polymers-13-04429-f005:**
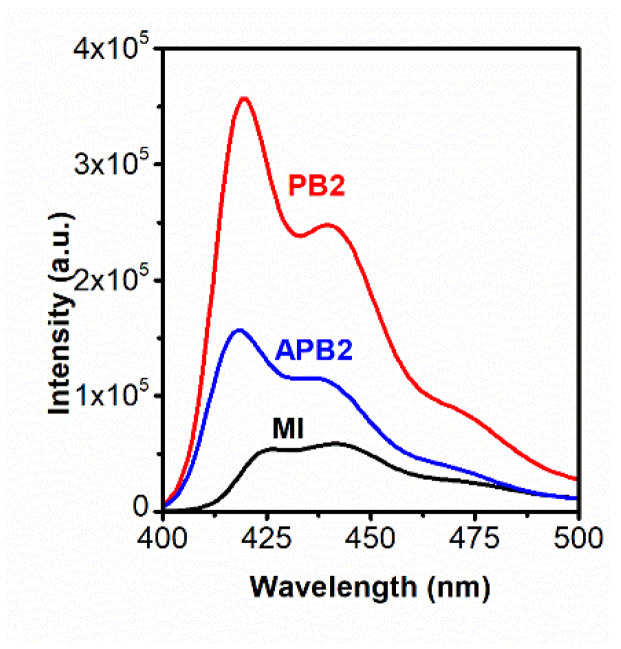
Fluorescence spectra of MI, PB2 and APB2.

**Figure 6 polymers-13-04429-f006:**
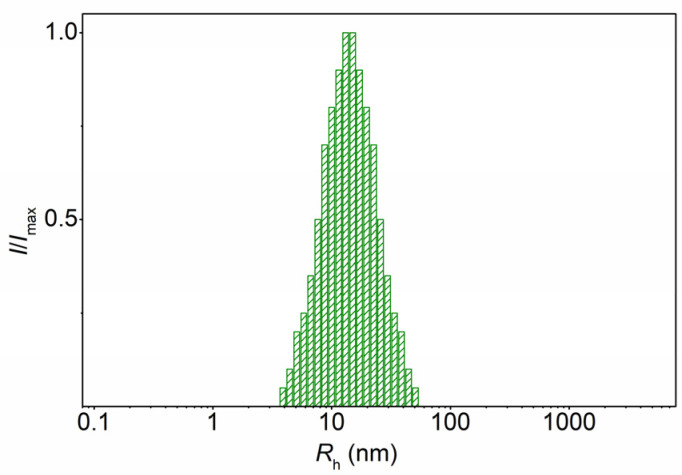
Hydrodynamic radii distribution for solution copolymer APB3 at concentration in ethanol *c* = 0.0035 g∙cm^−3^.

**Figure 7 polymers-13-04429-f007:**
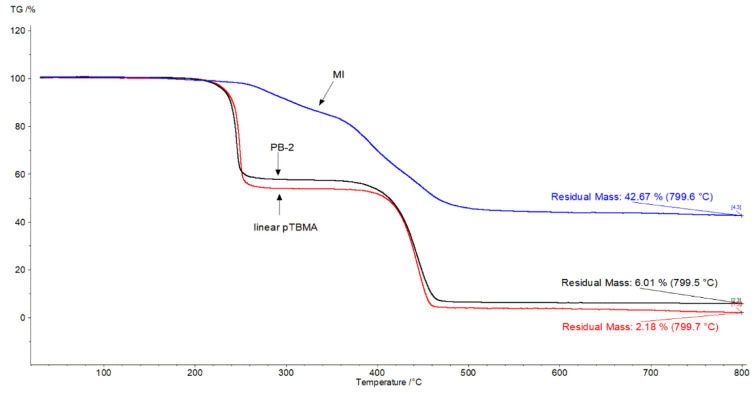
Thermogravimetric analysis data of MI, PB2 and linear PtBMA.

**Table 1 polymers-13-04429-t001:** Characteristics of solvents and solubility of the structural elements of molecular brushes.

Solvent	Solvent Characteristics			Solubility of the Structural Elements of Molecular Brushes		
	ρ, g·cm^−^^3^	η_0_, cP	*n* _0_	PF	PtBMA	PMAA
Chloroform	1.49	0.57	1.446	+	+	+
Ethanol	0.79	1.08	1.359	+/−	−	+
THF	0.89	0.46	1.405	+	+	−

**Table 2 polymers-13-04429-t002:** Conditions of syntheses and characteristics of PBs and MI.

Sample	MI:TBMA	τ ^4^, h	*A*^5^, %	*M*_n_^7^ × 10^–3^, g·mol^–1^	*M*_w_/*M*_n_^7^	*T*_g_^8^, °C
PB1	1:50 ^1^	24	85 ^6^	170	2.4	116.2
PB2	1:50 ^1^	4.0	67	140	2.1	113.9
PB3	1:100 ^1^	1.0	21	202	2.0	116.2
PB4	1:100 ^1,2^	1.33	37	178	2.4	116.2
PB5	1:200 ^3^	2.2	53	260	1.9	118.1
PB6	1:200 ^3^	2.8	68	290	1.8	118.4
PB7	1:200 ^3^	3.5	71	410	1.7	118.9
PB8	1:200 ^3^	4.2	76	510	1.8	119.5
MI				21.0	1.5	-

^1^ ATRP mechanism: molar ratio [MI]:[CuCl]:[dNbpy] = 1:1:3, [CuCl_2_] = 3 mol% of CuCl; anisole 33 vol% was used as a solvent; ^2^ no CuCl_2_ was added; ^3^ ATRP AGET mechanism: molar ratio [MI]:[CuCl_2_]:[Sn(EH)_2_]:[dNbpy] = 1:1:1:3, anisole 33 vol% was used as a solvent; ^4^ time, hour; ^5^ monomer conversion was determined using gas chromatography, %; ^6^ conversion was determined by weight method; ^7^ SEC was performed using a light scattering detector; ^8^ Glass transition temperature.

**Table 3 polymers-13-04429-t003:** Molar masses, hydrodynamic, structural conformational characteristics, and quantum yields of the studied PBs in chloroform.

Sample	*M*_n_^cal^ × 10^−3^, g·mol^−1^	*M*_w_^cal^ × 10^–3^, g·mol^−1^	*M*_w_^exp^ × 10^–3^, g·mol^−1^	*M*_w_^exp^/*M*_w_^cal^	*L*_s.ch._/Δ*L*	[η], cm^3^g^−1^	*R*_h-D_, nm	*A,* nm	*A*_0_ × 10^10^, erg·K^−1^mol^−1/3^	*d,* nm	QY
PB1	147	353	320	0.91	9.5	27	10	17	3.2	11	0.75
PB2	147	353	330	0.93	9.8	24	21	16	1.1	10	0.77
PB3	276	579	282	0.49	8.2	24	13	18	2.3	9	0.64
PB4	276	552	369	0.67	11.1	26	16	17	1.9	10	0.72
PB5	534	1014	375	0.37	11.3	30	17	18	2.2	11	0.73
PB6	534	1281	353	0.28	10.6	50	24	21	2.4	15	-
PB7	534	1281	579	0.45	18	41	30	21	1.3	14	-
PB8	534	1281	552	0.43	17	50	32	22	1.4	16	0.77

**Table 4 polymers-13-04429-t004:** Molar masses, hydrodynamic characteristics, and quantum yields of APBs.

Sample	*M*_w_ × 10^−3^, g·mol^−1^	*R*_h-m_, nm	*R_g_,* nm	*R_g_/R* _h_	[η], cm^3^g^−1^	*R*_h-m_, nm in Water	QY
APB1	479	16	37	2.3	33	17	0.68
APB2	500	20	44	2.2	35	22	0.64
APB3	715	14	40	2.9	31	15	0.71
APB4	368	12	37	3.1	35	16	-
APB5	700	19	73	3.8	35	20	0.72
APB6	1500	24	94	3.9	50	22	-
APB7	4000	31	129	4.1	49	32	-
APB8	720	24	41	1.7	61	30	0.69

## Data Availability

The data presented in this study are available on request from the corresponding author.

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
