# Peer review of "Novel Amphiphilic Polyfluorene-Graft-(Polymethacrylic Acid) Brushes: Synthesis, Conformation, and Self-Assembly"

_polymers, 2021, doi:10.3390/polym13244429_

Round 1

Reviewer 1 Report

Reference Report

Title:  Polymer brushes polyfluorene-graft-polymethacrylic acidby

Simonova et al.

Manuscript ID: polymers-1494419

Comments:

This work studied the characterization of polyfluorene polymer brushes with polymethacrylic acid side chains. The authors found that due to the brushes amphiphilic nature, they can form uni-molecular micelles with interaction with water, and therefore this new polymer brushes can potentially be used in drug delivery and bio-imaging. This work is comprehensive and thorough. I only have some minor comments:

  1. Introduction, L62: When mentioning polymer nanomaterials for drug delivery and bio-imaging, some significant references such as Siddique et al (Nanomaterials 2020;10:1700) should be included.
  2. Introduction: It is good to have a schematic diagram of the AFB with PF backbone and PMAA side chain.
  3. Materials and Methods: It is good to have a schematic diagram or flow chart showing how to fabricate the polymer step by step. Or a table can be added to summarize all subsections in Section 2. Moreover, there are too many subsections in Section 2.
  4. Figure 2: Error bars are missing in the plot.
  5. Table 1 in L285 should be “Table 2”. Also, please explain why there is no data for the Tg (glass transition temperature) column with PB6-8.
  6. Figure 3: The unit in the y-axis of the plot is missing. Please also label MI, PB1-4 as 3(A) and MI, PB5-8 as 3(B).
  7. Equation 1: Please confirm it should be Ms,ch or Ms,c in the text.
  8. Equation 6 is missing in the manuscript.
  9. Figure 5: Can the plot use a curve rather than histogram?
  10. Conclusions: Please discuss more about the potential applications of the AFB in bio-imaging and drug delivery. Please state the future work on this polymer.

Author Response

  1. Introduction, L62: When mentioning polymer nanomaterials for drug delivery and bio-imaging, some significant references such as Siddique et al (Nanomaterials 2020;10:1700) should be included.

Thank you for your suggestion. we are pleased to include this review in our cited literature. Line 68.

  1. Introduction: It is good to have a schematic diagram of the AFB with PF backbone and PMAA side chain.

Thank you for your remark. We have included a drawing of the polymer structure in the article. Line 64.

  1. Introduction: Materials and Methods: It is good to have a schematic diagram or flow chart showing how to fabricate the polymer step by step. Or a table can be added to summarize all subsections in Section 2. Moreover, there are too many subsections in Section 2.

Thank you for your comment. We have added the necessary information to Figure 2. In order not to confuse the reader, we decided not to add the flowchart. Moreover, we removed unnecessary subsections.

  1. Figure 2: Error bars are missing in the plot.

We added error bars in this figure 3 (numbering has been changed because we added new figure). Line 263.

  1. Table 1 in L285 should be “Table 2”. Also, please explain why there is no data for the Tg (glass transition temperature) column with PB6-8.

Thank you for your remark. We have made changes, line 268. We carried out additional measurements and entered the glass transition temperatures in the table.

  1. Figure 3: The unit in the y-axis of the plot is missing. Please also label MI, PB1-4 as 3(A) and MI, PB5-8 as 3(B).

Thanks. we have signed the dimension of the axis in the figure 4 and also labeled MI, PB1-4 as 4(A) and MI, PB5-8 as 4(B) (numbering has been changed because we added new figure). Line 290.

  1. Equation 1: Please confirm it should be Ms,ch or Ms,c in the text.

We confirmed that Equation 1: Ms,ch. We have made changes. Line 350.

  1. Equation 6 is missing in the manuscript

We have added the lost formula. Line 410

  1. Figure 5: Can the plot use a curve rather than histogram?

Thank you. Figure 6 (numbering has been changed because we added new figure). This is the most realistic depiction of the hydrodynamic size distribution.

  1. Conclusions: Please discuss more about the potential applications of the AFB in bio-imaging and drug delivery. Please state the future work on this polymer.

Thank you for your interest in our work. Finally, we have added information about upcoming studies. Lines 500-507.

Reviewer 2 Report

In this manuscript, the authors described the synthesis of amphiphilic molecular brushes with the polyfluorine backbone and PtBMA and PMAA side chains by radical polymerization. The structure and composition of synthesized molecular brushes were characterized using various methods.

  1. The title of this manuscript is confusing, some modification is needed.
  2. What is the purity of the chemical regents?
  3. In figure 1, the authors should label all of the compounds with numbers. Some reaction conditions are missing in the equation of synthesis P1 (phenylboronic acid pinacoline ester, p-bromoethoxy benzene).
  4. The authors only provided the 1H NMR of the intermediates, some of 13C NMR data should be provided to confirm the structure.
  5. Did the authors investigate the morphologies and amphiphilicity of the polymer?
  6. The attached supplementary material is empty.

Author Response

  1. The title of this manuscript is confusing, some modification is needed.

Thank you for your revision. We absolutely agree with you. We have changed the title of the article. «Novel amphiphilic polyfluorene-graft-(polymethacrylic acid) brushes: synthesis, conformation and self-assembly». Lines 2-3.

  1. What is the purity of the chemical regents?

This information was added in article. Lines 70-87.

  1. In figure 1, the authors should label all of the compounds with numbers. Some reaction conditions are missing in the equation of synthesis P1 (phenylboronic acid pinacoline ester, p-bromoethoxy benzene).

 We have added the required captions in Figure 2 (numbering has been changed because we added new figure). Line 243.

  1. The authors only provided the 1H NMR of the intermediates, some of 13C NMR data should be provided to confirm the structure.

We have added a spectrum 13С NMR description for the monomer. Lines 111-112.

The 13С NMR spectrum of the polymers was found to be not informative. We can¢t calculate functionalization degree of 2-bromo-isobutyrate groups of macroinitiator using 13С NMR spectrum therefore we used 1H NMR spectroscopy for this purpose.

We believe that the 1H spectra give us sufficient information about the structure of polymers.

We have obtained polyfluorenes by Suzuki polycondensation. Two monomers have been used in an equimolar ratio. Fragments of these monomers alternate in the polymer structure in accordance with the polycondensation mechanism. Therefore we suppose that 13С NMR spectra are not necessary in this case.

Figure 1 (in the uploaded file)

  1. Did the authors investigate the morphologies and amphiphilicity of the polymer?

Unfortunately, we have not investigated the morphology and amphiphilicity of these polymers yet.

The attached supplementary material is empty.

We are attaching this file in two formats

An error occurred while formatting the files. We apologize and are presenting supplementary material in two formats.

Round 2

Reviewer 1 Report

The authors answered my comments well with corresponding modifications in the revised manuscript.